# To What Extent Does Environmental Regulation Influence Emission Reduction? Evidence from Local and Neighboring Locations in China

**Jing Song [1], Mengyuan Li [2], Shaosong Wang [1],\*** and **Tao Ye [1]**

[1] School of Business, Macau University of Science and Technology, Macao 999078, China
[2] School of Economics, Shenzhen Polytechnic, Shenzhen 518055, China
\* Correspondence: sswang@must.edu.mo

**Abstract:** Setting environmental regulation policy is an important practice to reach the goal of ecological environmental protection. By establishing fixed effects and spatial spillover models, this paper examines the relationship between the environmental regulation of carbon emissions and the impact on spatial spillovers. The results of our analysis show that: 1. environmental regulation has significant inhibitory effects on carbon emissions, which is beneficial to carbon dioxide emission reduction, and 2. environmental regulation has a significant inhibitory impact on local carbon emissions but increases carbon emissions in neighboring locations. Therefore, in order to achieve the aim of carbon emission reduction, appropriate environmental regulation policies should be established, which, in turn, would provide better coordination of the carbon emission relationship between different regions. Meanwhile, environmental regulation plays an important role in protecting the environment. To strengthen environmental governance and promote the coordinated development of regional carbon emission reduction, we need to implement a top-level design of environmental regulation and build a market-oriented environmental regulation system.

**Keywords:** environmental regulation; carbon emissions; regional coordination

## 1. Introduction

Since the 21st century, the industrialization process of China has been continuously expanding and has been accompanied by a large amount of carbon dioxide emissions. According to the latest world energy statistics report, China's carbon dioxide emissions rose from 2.97 billion tons in 2001 to 9.5 billion tons in 2018. Currently, China is the largest carbon dioxide emitter [1] in the world. Carbon dioxide emissions lead to significant harm to people's health and ecosystems such as respiratory diseases and greenhouse effects [2]. To protect the environment and reduce carbon emissions, the Chinese government has formulated a series of environmental regulations, leading to the development of low-carbon industries. Moreover, the Chinese government had promised a 40–45% reduction in carbon emission intensity by 2020 compared with 2005. These environmental regulatory policies are helpful to control carbon dioxide emissions. However, they may lead to pollution relocation, which, in turn, enhances carbon emissions in neighboring areas. Therefore, regulators and researchers keep bringing to light the impact of 'local-neighboring' carbon dioxide emission reduction [3].

Previous studies find that environmental regulations have a direct impact on reducing local carbon emissions. Establishing environmental regulations is then considered an effective method to fix environmental issues and reduce carbon dioxide emissions [4]. However, the findings of the relationship between environmental regulation and carbon dioxide are not consistent [5]. Some studies argue that environmental regulation would reduce carbon emissions and that controlling environmental regulation plays an important role in improving ecological efficiency in Central and Western China [6]. Other studies,

however, indicate that environmental regulation could increase carbon emissions since the polices might lead to the accelerated development of fossil energy (the green paradox) and in turn, increase carbon emissions and air pollution [7]. Similarly, Werf and Maria (2012) argued that a lag in implementation and subsidies of alternative energy resources may lead to an increase in current energy consumption and carbon emissions [8]. As the existing findings are contradicting, this paper aims to contribute to the related literature by revealing a clearer picture of the link between environmental regulations and carbon emissions in China.

Most studies examining the impact of environmental regulation on carbon emissions generally ignore the fact that companies have adaptive options under strict environmental regulations, such as relocating heavily polluting factories to areas with weak environmental regulations [9]. Since local governments have different incentive policies for liquidity resources [10] and unequal economic and technological innovation, heavy pollution enterprises could relocate to evade the governance of environmental regulations and reduce environmental governance costs [11]. Therefore, the local environmental regulation may relocate pollution to nearby areas. There are only a limited number of studies exploring the effect of local environmental regulation on carbon dioxide emission reduction in neighboring areas [12]. To enrich this line of research, we established a fixed effect and spatial spillover effect model to test the impact of environmental regulation on local and adjacent carbon emissions.

Specifically, we tried to establish a link between environmental regulation and carbon emissions, and further test the different effects of local and neighboring areas. Our findings show that environmental regulations reduce $CO_2$ emissions locally but increase $CO_2$ emissions in the neighboring areas.

The remainder of the paper is organized as follows: Section 2 reviews the relevant literature, Section 3 describes the data and methodology, Section 4 provides empirical findings and discussions and conclusions are in Section 5.

## 2. Literature Review and Hypothesis Development

Carbon emissions have a significant impact on social progress and economic development. The existing literature on carbon emissions also mainly focuses on the following three aspects:

(1) Technological Progress. Grossman and Kruegert (1995) argue that the reduction of emissions mainly results from technological progress [13]. They developed the environmental Kuznets curve to reveal the inverted U-shaped curve relationship between economic development and environmental pollution. Based on the pollution shelter theory, Oates and Portney (2003) document that local governments would attract external investment by deregulation to achieve economic development [14]. In addition, Gorg and Greenaway (2004) believe that foreign technology spillover plays a positive role in carbon reduction [15].

(2) Industrial Structure. Using China's interprovincial panel data, Zhang et al. (2014) empirically analyze the relationship between industrial structure and carbon emissions and point out that secondary industry is the most important industry affecting carbon emissions [16]. They conclude that industrial upgrading would reduce the carbon emissions of cities. Moreover, Zhang et al. (2018) focus on the impact of industrial agglomeration on urban carbon emissions [17]. The empirical results show a significantly negative relationship between manufacturing agglomeration and urban carbon emissions. As a result, one of the crucial ways to reduce industrial carbon emissions is the improvement of enterprise agglomeration.

(3) Environmental Regulation. Hao et al. (2021) find that environmental regulation and FDI spillover effects have complementary effects on carbon emission technology [18]. Similarly, Chen et al. (2020) indicate a negative U-shaped relationship between environmental regulation and carbon emissions [19]. In addition, Whitmarsh et al. (2011) claim

that public participation in environmental regulation and voluntary environmental regulation could effectively reduce carbon emissions [20].

As mentioned above, the literature on carbon emissions mainly focuses at the government level (environmental regulation and industrial structure) and the internal aspect of enterprises (technological progress). However, most of them explore the influence path of environmental regulation on carbon emissions, but generally ignore the possible impact of environmental policies on the adjacent carbon dioxide emissions. Therefore, to fill in the blanks of the relevant research, this paper examines the impact of environmental regulation on carbon emissions through the construction of fixed-effect and space spillover models, and further analyzes the relationship between environmental regulation and neighbor-local carbon emission reduction.

## 3. Data and Methodology

### 3.1. Dependent Variable

We employed the environmental regulation and carbon emissions panel data from 30 provinces in China between 2006 and 2016. The data was obtained from China's energy statistical yearbook, China Statistical Yearbook and China Stock Market Accounting Research (CSMAR) Regional Economies Database. We calculated the carbon emissions from eight categories of fossil fuel in the manner recommended by IPCC (IPCC, 2007) as follows:

$$CO2E = \sum_{i=1}^{8} (CO_2)_i = \sum_{i=1}^{8} E_i \times NCV_i \times CEF_i \times COF_i \times 44/12 \tag{1}$$

where *CO2E* is the carbon emission (unit: ton), *i* is the type of fossil fuel, *E* is the consumption of fossil fuel, *NCV* is the net calorific value, *CEF* is the carbon content, *COF* is the carbon oxidation rate and 44 and 12 are the molecular weight of carbon dioxide and carbon, respectively. We then obtained the carbon emission coefficients by using the *NCV*, *CEF* and *COF*. The carbon emission coefficients of the eight fossil fuels are shown in Table 1.

**Table 1.** The $CO_2$ emissions indexes of eight fossil fuels.

| $CO_2$ Emission Sources | Coal | Coke | Crude Oil | Gasoline | Diesel Oil | Fuel Oil | Natural Gas | Kerosene |
|---|---|---|---|---|---|---|---|---|
| emission factor (tC/t) | 0.4925 | 0.7705 | 0.8187 | 0.7977 | 0.8461 | 0.8691 | 0.5896 | 0.8281 |

### 3.2. Explanation Variable

We examined the explanation variable of environmental regulation by calculating the comprehensive index of pollutant emissions (e.g., sulfur dioxide and soot) and the economic development in each province. We first standardized the emissions of pollutants per unit of economic output as follows: $DE_{ij}^s = [DE_{ij} - \min(DE_j)]/[\max(DE_j) - \min(DE_j)]$ where $DE_{ij}^s$ represents the standardized estimates of pollutants *j* in province *i*, while $DE_{ij}$ is the actual amount of pollutants *j* in province *i*; $\min(DE_j)$ and $\max(DE_j)$ are the actual minimum and maximum pollutants *j* in each province, respectively. Then, we set the adjustment parameters as follows: $W_j = DE_{ij}/\overline{DE_{ij}}$ where $DE_{ij}$ is the mean value of pollutant *j*. Finally, we computed the environmental regulation in each province as follows: $ER_{it} = \sum_1^j W_j DE_{ij}^S$.

### 3.3. Controlling Variables

We employed some other variables in this study as controlling variables since these factors could significantly influence the carbon emissions, including energy efficiency (ENS) [13], urbanization level (URBAN) [5], total population (TP) [14], fixed asset investment (PIFI) [15] and economic export-oriented level (TRADE) [16]. A full description of each variable is given in Table 2.

**Table 2.** Descriptive variable statistics.

| Variable Name | Variable Definitions | Obs | Mean | SD | Min | Max |
|---|---|---|---|---|---|---|
| perco2 | carbon dioxide emissions per capita (t) | 330 | 6.460 | 3.840 | 1.330 | 24.96 |
| CO2E | carbon dioxide emissions (million t) | 330 | 255.78 | 176.35 | 16.50 | 842.20 |
| ER | environmental regulation intensity | 330 | 0.470 | 0.86 | 0 | 7.20 |
| ENS | energy utilization efficiency (kg/yuan) | 330 | 1.060 | 0.49 | 0.23 | 3.12 |
| URBAN | urbanization level | 330 | 50.59 | 14.34 | 26.28 | 89.60 |
| TP | year-end total population (thousands of people) | 330 | 4400 | 2646 | 539 | 10,724 |
| PIFI | the proportion of fixed-asset investment in GDP | 330 | 0.62 | 0.19 | 0.25 | 1.24 |
| TRADE | share of total imports and exports in GDP | 330 | 0.05 | 0.06 | 0.01 | 0.24 |

*3.4. Models*

As discussed above, our primary goal was to investigate the relationship between environmental regulation and carbon emissions using panel data. We constructed the following model to address the research goal:

$$DE_{it} = \beta_0 + \beta_1 ER_{it} + \sum \beta_i Control_{it} + \gamma_i + \delta_t + \mu_{it} \qquad (2)$$

where

$DE_{it}$ is the amount of carbon dioxide emissions

$ER_{it}$ is the environmental regulation of city $i$ in year $t$

$Control_{it}$ is the series of controlling variables

$\gamma_i$ is the province (individual) fixed effect

$\delta_t$ is the time fixed effect.

The second research goal aimed to assess the effect of spatial spillover of environmental regulation on carbon emissions in a particular province. Our second model is as follows:

$$DE_{it} = \beta_0 + \rho_0 W y_{it} + \beta_1 ER_{it} + \sum \beta_i Control_{it} + \theta_1 WER_{it} + \sum \theta_i WControl_{it} + \mu_{it} \qquad (3)$$

where

$\rho_0$ is the spatial lag coefficient

$\theta_i$ is the spatial exchange term coefficient

W is the spatial weight matrix

We used the 0–1 adjacency method to construct the spatial weight matrix.

## 4. Empirical Findings

According to the test results of the Hausman panel regression model (Table 3), we can see that chi2(7) equals to 38.35 and 44.59, and the *p*-value is less than 0.01. Therefore, we reject the hypothesis and a fixed effect model is selected in this study.

*4.1. Main Effects Regression Analysis*

The results of estimating the carbon emission models, lnperco2 (logarithm value of $CO_2$ per capita) and lnCO2E (logarithm value of $CO_2$), are shown in Table 4. After controlling for energy efficiency, urbanization levels, total population, fixed asset investment and economic export-oriented levels, we find that environmental regulations significantly reduce carbon emissions (coefficient: $-0.0356$ (lnperco2) and $-0.0429$ (lnCO2E); t-value: $-2.8998$ (lnperco2) and $-2.9062$ (lnCO2E)). The results indicate that the strengthening of environmental regulation could promote $CO_2$ emission reduction in various provinces, which is in line with [13]. In addition, with lnperco2 as the explanation variable, the coefficients of total population size (TP) and the economic export-oriented level (TRADE) are both significantly negative, indicating that an increase in population size and the development of international trade could reduce the per capita $CO_2$ emissions. Moreover, the relationship between energy efficiency (ENS) and carbon emissions is significantly negative. This indicates that under stricter environmental regulation standards, the costs

would be increased if companies fail to improve the production methods. As a result, to reduce the production costs, companies need to improve their green technology innovations to increase energy efficiency and enhance their capabilities in emission reduction [17].

**Table 3.** Hausman panel regression result.

| | (1) | (2) | (3) | (4) |
|---|---|---|---|---|
| | FE | RE | FE | RE |
| | lnperco2 | lnperco2 | lnCO2E | lnCO2E |
| ER | −0.0356 *** | −0.0368 *** | −0.0429 *** | −0.0457 *** |
| | (−4.3231) | (−4.2966) | (−4.3690) | (−4.4276) |
| ENS | −0.4008 *** | −0.4450 *** | −0.3895 *** | −0.4546 *** |
| | (−9.7829) | (−10.6702) | (−7.9736) | (−9.0493) |
| URBAN | 0.0109 *** | 0.0170 *** | 0.0132 *** | 0.0199 *** |
| | (3.6296) | (7.3135) | (3.7107) | (7.1626) |
| PIFI | 0.1861 *** | 0.2320 *** | 0.2835 *** | 0.3616 *** |
| | (2.8708) | (3.6683) | (3.6681) | (4.7466) |
| TRADE | −1.1829 *** | −0.7822 * | −1.1291 ** | −0.6977 |
| | (−2.7291) | (−1.8723) | (−2.1847) | (−1.3881) |
| lnTP | −0.6458 *** | −0.1256 ** | 0.0935 | 0.8265 *** |
| | (−5.0425) | (−2.3612) | (0.6121) | (13.1297) |
| Constant | 6.6062 *** | 2.1019 *** | 3.7136 *** | −2.5439 *** |
| | (6.0390) | (4.4889) | (2.8470) | (−4.5886) |
| N | 330 | 330 | 330 | 330 |
| individual effect | yes | yes | individual effect | yes |
| year effect | yes | yes | year effect | yes |
| $R^2$ | 0.9017 | | 0.9141 | |
| Hausman-Test | chi2(7) = 38.35 prob > chi2 = 0.000 | | chi2(7) = 44.59 prob > chi2 = 0.000 | |

(1) *t*-statistics are in parentheses. (2) * $p < 0.1$, ** $p < 0.05$, *** $p < 0.01$.

**Table 4.** Results of main effects regression analysis.

| | lnperco2 | lnCO2E |
|---|---|---|
| | (1) | (2) |
| ER | −0.0356 *** | −0.0429 *** |
| | (−2.8998) | (−2.9062) |
| ENS | −0.4008 *** | −0.3895 *** |
| | (−3.9826) | (−3.6464) |
| URBAN | 0.0109 ** | 0.0132 ** |
| | (2.0532) | (2.4647) |
| PIFI | 0.1861 | 0.2835 ** |
| | (1.4808) | (2.1892) |
| TRADE | −1.1829 ** | −1.1291 * |
| | (−2.6109) | (−1.9561) |
| lnTP | −0.6458 ** | 0.0935 |
| | (−2.7144) | (0.3318) |
| Constant | 6.6062 *** | 3.7136 |
| | (3.4296) | (1.6266) |
| N | 330 | 330 |
| individual effect | yes | yes |
| year effect | yes | yes |
| Adj. $R^2$ | 0.8967 | 0.9097 |

(1) The *t*-statistics, reported in parentheses, are obtained after considering Driscoll and Kraay standard errors. (2) * $p < 0.1$, ** $p < 0.05$ and *** $p < 0.01$.

### 4.2. Spatial Spillover Effect Analysis

We further investigate whether regional environmental regulation (RER) has spatial spillover effects on $CO_2$ emissions. Specifically, we adapt the spatial Durbin model (SDM) to analyze the spillover effects on neighboring locations [18]. The results of the spatial panel Durbin model test are shown in Table 5. We find that environmental regulations

significantly reduce local carbon emissions but increase carbon emissions in neighboring areas. One potential explanation is that the different intensity of environmental regulations among provinces may induce the relocation of polluting industries. Relevant companies moving to neighboring provinces leads to higher neighboring carbon emissions. In addition, the coefficient of the urbanization level (URBAN) is significantly positive, indicating that carbon emissions increase with urbanization levels. Furthermore, we observe a significantly negative impact of international trade (TRADE) on carbon emissions, demonstrating that the development of international trade is conducive to the reduction of carbon emissions. We also find that energy utilization efficiency exhibits a significant inhibitory effect on $CO_2$ emissions in local areas, but not in neighboring areas. This phenomenon shows that governments may encourage enterprises to improve energy utilization efficiency through policies, such as technological innovation and the purchase of cleaner production equipment, which eventually reduces carbon emissions.

**Table 5.** Results of spatial spillover effect analysis.

| | | lnperco2 | lnCO2E |
|---|---|---|---|
| | | **(1)** | **(2)** |
| Local effect | ER | −0.0206 *** | −0.0260 *** |
| | | (0.0075) | (0.0090) |
| | ENS | −0.3655 *** | −0.3430 *** |
| | | (0.0434) | (0.0520) |
| | URBAN | 0.0145 *** | 0.0174 *** |
| | | (0.0029) | (0.0035) |
| | lnTP | −0.7144 *** | 0.1433 |
| | | (0.1510) | (0.1812) |
| | PIFI | 0.1414 ** | 0.2734 *** |
| | | (0.0618) | (0.0740) |
| | TRADE | −1.2534 *** | −1.2583 ** |
| | | (0.4096) | (0.4915) |
| Neighbor effect | ER | 0.0532 *** | 0.0625 *** |
| | | (0.0191) | (0.0229) |
| | ENS | 0.0505 | 0.0815 |
| | | (0.0939) | (0.1109) |
| | URBAN | −0.0291 *** | −0.0310 *** |
| | | (0.0059) | (0.0071) |
| | lnTP | −0.4409 | −0.7852 ** |
| | | (0.2984) | (0.3423) |
| | PIFI | 0.5098 *** | 0.5026 *** |
| | | (0.1211) | (0.1454) |
| | TRADE | −1.4161 * | −2.0924 ** |
| | | (0.7955) | (0.9550) |
| | spatial-rho | −0.1780 ** | −0.2321 *** |
| | | (0.0814) | (0.0847) |
| | sigma2_e | 0.0035 *** | 0.0050 *** |
| | | (0.0003) | (0.0004) |

(1) The *t*-statistics, reported in parentheses, are obtained after considering Driscoll and Kraay standard errors. (2) * $p < 0.1$, ** $p < 0.05$ and *** $p < 0.01$.

### 4.3. Robust Analysis

In order to prove our results are convincing and reliable, we need to test the robustness of the model. Referring to Gormley and Matsa et al. (2014) and Xu (2018), we use the high-dimensional fixed effect model to estimate the regression parameters to test the robustness of the regression results [21,22]. It can be seen from the results in Table 6 that the regression coefficient symbol of environmental regulation as an explanatory variable is consistent with the previous estimators, and the results are not very different, indicating our results are robust.

**Table 6.** Robustness test based on high-dimensional fixed effect.

|  | (1) | (2) |
|---|---|---|
|  | lnperco2 | lnCO2E |
| ER | −0.0356 *** | −0.0429 *** |
|  | (−2.8998) | (−2.9062) |
| ENS | −0.4008 *** | −0.3895 *** |
|  | (−3.9826) | (−3.6464) |
| URBAN | 0.0109 ** | 0.0132 ** |
|  | (2.0532) | (2.4647) |
| PIFI | 0.1861 | 0.2835 ** |
|  | (1.4808) | (2.1892) |
| TRADE | −1.1829 ** | −1.1291 * |
|  | (−2.6109) | (−1.9561) |
| lnTP | −0.6458 ** | 0.0935 |
|  | (−2.7144) | (0.3318) |
| Constant | 7.0118 *** | 4.1708 * |
|  | (3.6418) | (1.8276) |
| N | 330 | 330 |
| individual effect | yes | yes |
| year effect | yes | yes |
| Adj. $R^2$ | 0.9744 | 0.9889 |

(1) $t$-statistics are in parentheses. (2) * $p < 0.1$, ** $p < 0.05$ and *** $p < 0.01$.

Furthermore, the two-stage least squares (2SLS) method is used in this study to reduce the reverse causal issue of the relationship between environmental regulation and carbon emissions. In this paper, the lagged environmental regulation (L.ER) is used as an instrumental variable. It can be seen from Table 7 that the variable L.ER passes the insufficient identification test and the weak instrumental variable test. In addition, through the results of the first stage, it can be found that L.ER and ER are positively correlated at the significant level of 1%, which jointly confirms the reasoning of the instrumental variable selected in this paper. Moreover, from the regression results of the second stage (columns 3 and 4 of Table 7), the coefficients of ER are significant at the level of 1%, that is, the higher the degree of environmental regulation, the more conducive each province is to reducing carbon emission levels.

**Table 7.** 2SLS regression results.

|  | (1) | (2) | (3) |
|---|---|---|---|
|  | First Stage | Second Stage | |
| Variables | ER | lnperco2 | lnCO2E |
| ER |  | −0.2803 *** | −0.3447 *** |
|  |  | (−4.8627) | (−5.1988) |
| L.ER | 0.4186 *** |  |  |
|  | (7.2416) |  |  |
| ENS | −0.2252 ** | −0.4878 *** | −0.4954 *** |
|  | (−2.2710) | (−10.8224) | (−9.5559) |
| URBAN | 0.0011 | 0.0112 *** | 0.0131 *** |
|  | (0.1484) | (3.4733) | (3.5413) |
| PIFI | 0.1105 | 0.1925 *** | 0.2929 *** |
|  | (0.6896) | (2.8380) | (3.7548) |
| TRADE | −0.2362 | −1.1817 *** | −1.1336 ** |
|  | (−0.2245) | (−2.6818) | (−2.2366) |
| lnTP | 0.6584 * | −0.3718 ** | 0.4550 *** |
|  | (1.9678) | (−2.4719) | (2.6298) |
| Constant | −4.7877 * | 4.6881 *** | 1.0738 |
|  | (−1.6510) | (3.6884) | (0.7345) |
| Anderson canon. corr. LM statistic |  | 51.172[0.0000] | |
| Cragg–Donald Wald F statistic |  | 52.441{16.38} | |
| N | 300 | 300 | 300 |
| individual effect | yes | yes | yes |
| year effect | yes | yes | yes |
| Adj. $R^2$ |  | 0.978 | 0.991 |

(1) $t$-statistics in parentheses. (2) * $p < 0.1$, ** $p < 0.05$ and *** $p < 0.01$.

This table reports the estimated regression coefficients of 2SLS. This paper uses the lag period of the independent variable as our instrumental variable. All regression included control variables; [] is *p* value; {} is the critical value of the Stock–Yogo weak instrument variable at 10% significance level. The original assumption of the Anderson canon. corr. LM test is that the identification of instrumental variables is insufficient, and the value in [] is *p*. The original assumption of the Cragg–Donald Wald F test is that the instrumental variable is a weak instrumental variable, and the critical value of the Stock–Yogo weak instrumental variable at 10% significance level is in {}.

## 5. Conclusions

Previous studies have examined the impact of environmental regulation on carbon emissions; however, their findings were inconclusive [20]. Some researchers reported a positive link between environmental regulation and carbon emissions [6,19], while others argued oppositely [7,21,22]. However, most existing studies generally ignored the different impacts of certain environmental regulations on carbon emissions in local and neighboring areas [23–26].

Our study aims to investigate the impact of environmental regulations on carbon emissions in China, as well as the different impacts in local and neighboring areas [27]. To achieve this, we established a fixed effect model with panel data from 2006 to 2016. Our model controls for energy efficiency, urbanization levels, total population, fixed asset investment and economic export-oriented levels. With comprehensive analysis, we find a significantly negative impact of regulation on carbon emissions. We also observe that the impact is significantly different in the local and neighboring areas. Our findings suggest that we need to shed more light on the regional coordination of environmental regulation policies. We believe our research may be of substantial value for the Chinese government to coordinate environmental regulations and carbon emission reduction decisions [28,29].

Our research, however, is subject to a few limitations: Firstly, besides the indicators of other quantitative measurements of environmental regulation which can be applied to analyze the evolutionary features of the carbon emissions, similar studies can be conducted at the provincial level to check the effectiveness of environmental regulations. In addition, future study could be expanded into other economic regions, such as BRICS, to reveal their features of carbon emissions. Moreover, this paper does not consider urban-level heterogeneity. It would be very meaningful if future studies considered the relationship between the urban level and the variables of an urban economic scale, population scale and industrial scale.

**Author Contributions:** Writing—original draft, J.S.; Writing—review & editing, M.L., S.W. and T.Y. All authors have read and agreed to the published version of the manuscript.

**Funding:** This research was funded by Research and Technology Administration Office, Macau University of Science and Technology, grant number [FRG-16-008-MSB].

**Institutional Review Board Statement:** Not applicable.

**Conflicts of Interest:** The authors declare no conflict of interest.

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
