# Peer review of "To What Extent Does Environmental Regulation Influence Emission Reduction? Evidence from Local and Neighboring Locations in China"

_sustainability, doi:10.3390/su14159714_

Round 1
Reviewer 1 Report
After reading the reviewed article, I can say the following:
1. This paper examines the relatonship between environmental regulations on carbon emissions and the impact on spatial spillovers. The data relates to CO2 emissions and comes from 30 provinces in the China between 2006 and 2016. The model controls for energy efficiency, level of urbanization, capital assets investment, and economic export-oriented level.
2. The article is well written and understandable to the readers.
3. The abstract of the article should be expanded; it is too short and does not fully reflect the content of the article.
4. There are incorrect entries of numerical values in the table 2 - numerical values starting with four digits should be written with the comma every three digits.
5. Minor ordering remarks have been made in the text of the article in the mode track changes.
6. The amanded article may be published in the Suastaiability journal.

Author Response
Thank you for review’s hard work, and thank you very much for review’s valuable comments. Please see the attachment

Reviewer 2 Report
This research work was previously revised and improved. It is not a classical experimental research work but not only bibliographic revision is done and a mathematical analysis let the authors to reach their conclusions. In consequence, it is an interesting research work that show new and interesting conclusions about regulations and emission reductions.
Author Response
Thank you for review’s hard work, and thank you very much for review’s valuable comments.